# Riemannian optimization of isometric tensor networks

**Markus Hauru**$^\star$**, Maarten Van Damme and Jutho Haegeman**

Department of Physics and Astronomy, Ghent University, Gent, Belgium

$\star$ markus@mhauru.org

## Abstract

Several tensor networks are built of isometric tensors, i.e. tensors satisfying $W^\dagger W = \mathbb{1}$. Prominent examples include matrix product states (MPS) in canonical form, the multiscale entanglement renormalization ansatz (MERA), and quantum circuits in general, such as those needed in state preparation and quantum variational eigensolvers. We show how gradient-based optimization methods on Riemannian manifolds can be used to optimize tensor networks of isometries to represent e.g. ground states of 1D quantum Hamiltonians. We discuss the geometry of Grassmann and Stiefel manifolds, the Riemannian manifolds of isometric tensors, and review how state-of-the-art optimization methods like nonlinear conjugate gradient and quasi-Newton algorithms can be implemented in this context. We apply these methods in the context of infinite MPS and MERA, and show benchmark results in which they outperform the best previously-known optimization methods, which are tailor-made for those specific variational classes. We also provide open-source implementations of our algorithms.

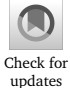

# 1  Introduction

Tensor networks can be used to efficiently represent vectors and operators in very large tensor product spaces, assuming they have a restricted structure of correlations. This makes them well-suited as ansätze for ground states of local quantum Hamiltonians and other quantum states with limited entanglement [1–3]; as compact representations of partition functions of large systems in classical statistical mechanics [4,5]; and as representations of tensors of various kinds in other applications [6], such as machine learning [7,8]. Many tensor networks have constraints applied to their tensors, most common of them being the requirement of isometricity, i.e. the property that $W^\dagger W = \mathbb{1}$ when the tensor $W$ is interpreted as a linear map from the tensor product space associated with a subset of its indices to the space associated with the complementary set of indices. This constraint arises from removing redundant gauge freedom from the network in the case of canonical forms of matrix product states (MPS) [2] and tree tensor networks (TTN) [9], but is inherent in the definition of the multiscale entanglement renormalization ansatz (MERA) [3]. Even for projected entangled-pair states (PEPS) [1,10], where an isometry constraint does not arise naturally, it might be interesting to consider the restricted set with isometric tensors, as this simplifies certain calculations [11–13]. Furthermore, tensor networks constructed from isometric tensors are equivalent to quantum circuits that could potentially be implemented on a quantum computer, and have attracted recent attention from this point of view [14–17].

To find a tensor network approximation of an unknown state of interest, e.g. a ground state of some local Hamiltonian, the variational principle is invoked, i.e. the ground state approximation is identified with the point on the tensor network manifold that minimizes the energy. The first algorithm for finding such an approximation was the density matrix renormalization group (DMRG) [18], which optimizes the energy over the set of MPS (although the MPS structure was only implicit in the original formulation of DMRG). The one-site DMRG algorithm in particular optimizes each tensor in turn, iterating the procedure until convergence, a technique known as alternating least squares optimization. A similar alternating optimization strategy is also the basis for the standard energy minimization algorithm for MERA [19], which we refer to as the Evenbly-Vidal algorithm, although the local problem is in this case solved differently in order to respect the isometry condition. Another paradigm for finding minimal energy tensor networks is based on the idea of imaginary time evolution, using either Trotter decompositions [20,21] or the time-dependent variational principle (TDVP) [22,23]. Trotter-based imaginary time evolution has been the prevailing algorithm for the optimization of infinite PEPS until recently [24,25]. In the context of optimizing unitary or isometric tensor networks, yet another strategy is based on flow equations, as proposed in Ref. 26. Also in the context of quantum computational tasks, classical optimization of the unitary gates in the quantum circuit with respect to a given cost function is often required, as e.g. in Ref. 17.

Well-known gradient-based algorithms for nonlinear optimization have not received a great deal of attention for the optimization of tensor networks, likely due to the astounding efficiency of the DMRG algorithm for the case of MPS. Promising results for using the standard (i.e. Euclidean) version of the nonlinear conjugate gradient algorithm were reported for translation-invariant MPS [27] and PEPS [25] in the thermodynamic limit. In this manuscript, we propose

to use the well-established Riemannian generalization of the nonlinear conjugate gradient and quasi-Newton algorithms to optimize over manifolds of isometric tensor networks. We furthermore construct a specific preconditioner for these algorithms, derived from the Hilbert space geometry of the tensor network manifold, and show that the resulting methods can outperform tailor-made optimization algorithms, such as the Evenbly-Vidal algorithm for MERA and the variational uniform MPS (VUMPS) algorithm [28] for infinite MPS.

This manuscript is structured as follows: Section 2 provides an overview of the Riemannian geometry of complex Grassmann and Stiefel manifolds, the manifolds of isometric matrices and tensors. In Section 3, we briefly review the basics of Riemannian extensions of gradient-based optimization methods such as the gradient descent, nonlinear conjugate gradient and quasi-Newton algorithms, and discuss the role of preconditioners in this setting. In Sections 4 and 5, we show how these methods can be applied in the context of MERA and MPS, respectively, and demonstrate how they outperform previous methods in many situations. Section 6 provides some further discussion and an outlook.

The algorithms presented below are available in open source software packages written in the scientific programming language Julia [29]. The most high-level and user-facing packages are MPSKit.jl [30] and MERAKit.jl [31]. The ancillary files in arxiv.org/src/2007.03638 include scripts that use these packages to reproduce all the benchmark results that we show.

## 2  Riemannian geometry of isometric tensors

Throughout this section, we focus on a single isometric matrix $W$ that fulfills $W^\dagger W = \mathbb{1}$. This could for instance be an isometry or disentangler of a MERA, with its top and bottom indices combined to single matrix indices, or an MPS tensor in left or right canonical form.

In contrast to most literature in numerical optimization, we focus on complex isometric matrices. Isometric matrices of a given size $n \times p$ form a manifold, called the Stiefel manifold, that can be naturally embedded in the Euclidean vector space $\mathbb{C}^{n \times p}$ of general complex $n \times p$ matrices:

$$\mathrm{St}(n, p) = \{ W \in \mathbb{C}^{n \times p} \mid W^\dagger W = \mathbb{1} \}, \tag{1}$$

where we have assumed the necessary condition $n \geq p$. The case $n = p$ yields the manifold of unitary matrices $\mathrm{U}(n)$, which is thus included as special case. For instance, for the isometries of a ternary MERA with bond dimension $D$, $n = D^3$ and $p = D$, whereas the corresponding disentanglers have $n = p = D^2$. In the case of left or right canonical MPS with physical dimension $d$ and bond dimension $D$, $n = dD$ and $p = D$. The isometry constraint imposes $p^2$ independent real-valued constraints and thus $\mathrm{St}(n, p)$ is a real manifold of dimension $(2n - p)p$. Note that as the isometry constraint is not holomorphic, $\mathrm{St}(n, p)$ cannot be understood as a complex manifold, and its tangent space cannot be given the structure of a complex subspace of $\mathbb{C}^{n \times p}$, a point to which we return below.

In many situations, what is of interest is not the exact isometry $W$ itself, but rather the subspace which it defines by the span of its $p$ columns. In those cases, one should identify $W$ with $WU$, where $U$ can be an arbitrary $p \times p$ unitary, and consider the equivalence class $[W] = \{WU \mid U \in \mathrm{U}(p)\}$. In a tensor network, this happens whenever the columns of $W$ correspond to a single virtual index, in which case a gauge transformation $U$ can be applied to it, while $U^\dagger$ can be absorbed into the leg of the tensor to which $W$ is connected. The manifold of such equivalence classes of isometric tensors $[W]$ is a quotient manifold known as the Grassmann manifold $\mathrm{Gr}(n, p) = \mathrm{St}(n, p)/\mathrm{U}(p)$.

While $\mathrm{Gr}(n, p)$ is here defined as the quotient manifold of two manifolds without complex structure, $\mathrm{Gr}(n, p)$ is itself a proper complex manifold with complex dimension $(n - p)p$, or

equivalently, real dimension $2(n-p)p$. This can be understood by noticing that the isometry condition is not necessary to define a subspace, so that $\mathrm{Gr}(n,p)$ can also be defined as $\mathrm{Gr}(n,p) = \mathrm{GL}(\mathbb{C},n)/(\mathrm{GL}(\mathbb{C},p) \times \mathrm{GL}(\mathbb{C},n-p))$, with $\mathrm{GL}(\mathbb{C},n)$ the general linear group of invertible complex $n \times n$ matrices. In fact, $\mathrm{Gr}(n,p)$ can then be given the structure of a Kähler manifold, which can be important when studying time evolution [22]. In contrast, optimization of real-valued functions on a manifold is only concerned with the Riemannian structure (and not with possible complex, symplectic, or Kähler structures), for which only the structure as real manifold is relevant, as we make more explicit below. Throughout the remainder of this manuscript, we will denote elements from Grassmann manifolds using a single representative $W$ of the corresponding equivalence class $[W]$, and assume that $W$ is isometric.

We briefly review the basic properties of Grassmann and Stiefel manifolds which are required to apply gradient-based optimization methods. For a more thorough introduction to the properties of Grassmann and Stiefel manifolds, see for instance Refs. 32, 33. Note though, that these references only consider real-valued matrices, whereas we review here the complex case.

## 2.1 Tangent vectors

For an isometric matrix $W \in \mathrm{St}(n,p)$, the tangent space at $W$ consists of all matrices $X$ for which $W^\dagger X$ is skew-hermitian. In other words,

$$X = WA + W_\perp B, \text{ where } A = -A^\dagger. \tag{2}$$

Here $W_\perp$ is a $n \times (n-p)$ isometric matrix such that $WW^\dagger + W_\perp W_\perp^\dagger = \mathbb{1}$, i.e. it is a unitary completion of $W$ (which is not unique). $B$ is an arbitrary $(n-p) \times p$ matrix. The skew-hermiticity condition on $A$ implies that the tangent space only allows for linear combinations with real-valued scalar coefficients, i.e. it is a vector space over $\mathbb{R}$, as mentioned above. Because optimization algorithms are formulated using only real-valued linear combinations of tangent vectors, this does not pose any restriction.

For a point on the Grassmann manifold represented by $W$, we can use the unitary gauge freedom to impose that tangent vectors satisfy the holomorphic condition $W^\dagger X = 0$. This amounts to restricting to tangent vectors with $A = 0$, and thus the tangent vectors on a Grassmann manifold can be parameterized as

$$X = W_\perp B, \tag{3}$$

with $B$ again being an arbitrary $(n-p) \times p$ matrix. Note that the $A = 0$ condition is preserved under complex linear combinations, as one would expect given the complex structure of $\mathrm{Gr}(n,p)$.

In both cases, Stiefel and Grassmann, we denote the tangent space at $W$ by $T_W$, to which we can append the manifold if we want to distinguish explicitly between the two cases.

## 2.2 Metric

Implicit in most gradient methods is the idea to use the partial derivatives of the cost function, which constitute a dual vector in the cotangent space, as a direction (i.e. a tangent vector) along which to update the state. This works fine if one assumes to be working in Euclidean space, but otherwise requires a metric. A natural metric for $T_W$, regardless of whether we are on a Stiefel or Grassmann manifold, is the Euclidean metric $g_W(X,Y) = \mathrm{Re}\,\mathrm{Tr}[X^\dagger Y]$, i.e. the real part of the Frobenius inner product in the embedding space $\mathbb{C}^{n \times p}$. Note that the real part of the inner product of a complex space defines a metric (a real symmetric bilinear), whereas the imaginary part defines a symplectic form. While a general metric depends on the base point $W$, for $g_W$ this dependence is not explicit.

Another natural metric for the Stiefel manifold is given by what is known as canonical metric, for which we refer to Ref. 32. In this manuscript we use the Euclidean $g_W$, as we found little difference between the two choices in our simulations, and the Euclidean metric is more closely related to the Hilbert space inner product and the preconditioning schemes for the tensor networks that we consider in later sections.

A metric allows one to map cotangent vectors to tangent vectors. In a case like ours, where the manifold is embedded in a Euclidean space, it more generally allows one to construct an orthogonal projection from the embedding space to the tangent space. For a given complex matrix $D \in \mathbb{C}^{n \times p}$, we define its orthogonal projection onto $T_W$ as the tangent vector $G$ for which $g_W(G, X) = \operatorname{Re} \operatorname{Tr}[D^\dagger X]$, for all $X \in T_W$. The solution for this projection is

$$G = D - \frac{1}{2} W(W^\dagger D + D^\dagger W) \qquad\qquad \text{if } W \in \operatorname{St}(n, p), \tag{4}$$

$$G = D - W W^\dagger D \qquad\qquad\qquad\qquad \text{if } W \in \operatorname{Gr}(n, p). \tag{5}$$

$D \mapsto G$ is a complex linear map for the Grassmann manifold, but only real linear for the Stiefel manifold. Although the names $D$ and $G$ purposefully refer to derivatives and gradients, note that Eqs. (4) and (5) can be used to project any arbitrary matrix from $\mathbb{C}^{n \times p}$ onto the tangent space $T_W$.

## 2.3 Gradients, retraction, and transport

For gradient optimization of a cost function $C(W)$, we can first compute the partial derivatives

$$D_{ij} = \frac{\partial C}{\partial \operatorname{Re} W_{ij}} + i \frac{\partial C}{\partial \operatorname{Im} W_{ij}} = 2 \frac{\partial C}{\partial W_{ij}^*}, \tag{6}$$

without taking the isometry condition into account. The complex linear combination here is chosen such that

$$\left. \frac{\mathrm{d} C(W + \epsilon X)}{\mathrm{d} \epsilon} \right|_{\epsilon = 0} = \operatorname{Re} \operatorname{Tr}[D^\dagger X], \quad \forall X \in \mathbb{C}^{n \times p} \tag{7}$$

(assuming that the cost-function can meaningfully be extended or continued to non-isometric matrices in such a way that the above derivative is well defined). Projecting $D$ onto the tangent space with Eq. (4) or (5) yields $G$, which is the tangent vector such that

$$g_W(G, X) = \left. \frac{\mathrm{d} C(W + \epsilon X)}{\mathrm{d} \epsilon} \right|_{\epsilon = 0}, \quad \forall X \in T_W. \tag{8}$$

$G$ will henceforth be referred to as the *gradient* of $C$.

This brings us to the next point, which is that we would often like to change our isometry $W$ by moving in the direction of a tangent vector $X \in T_W$, but $W + \epsilon X$ will only respect the isometry condition up to first order in $\epsilon$. To travel further in the direction of $X$ while staying on the manifold, Riemannian optimization algorithms employ the concept of *retraction*. A retraction $R_W(X, \alpha)$ is a curve parameterized by $\alpha \in \mathbb{R}$, an initial point $W$ such that $R_W(X, 0) = W$, and initial direction $X \in T_W$ such that $\frac{\partial}{\partial \alpha} R_W(X, \alpha)|_{\alpha = 0} = X$, that lies exactly within the manifold for all values of $\alpha$ in some interval containing $\alpha = 0$ (preferably $\alpha \in \mathbb{R}^+$).

For both Stiefel and Grassmann manifolds, several retraction functions exist, even if we impose the requirement that we must be able to numerically compute them efficiently. One natural choice to consider are geodesics, since the notion of retraction can be seen as a generalization thereof. Given a tangent vector $X = WA + W_\perp B$, the retraction

$$R_W(X, \alpha) = e^{\alpha Q_X} W, \quad \text{where } Q_X = \begin{bmatrix} W & W_\perp \end{bmatrix} \begin{bmatrix} A & -B^\dagger \\ B & 0 \end{bmatrix} \begin{bmatrix} W^\dagger \\ W_\perp^\dagger \end{bmatrix}, \tag{9}$$

is indeed a geodesic for the Grassmann manifold (where $A = 0$), but is not a geodesic with respect to the Euclidean metric for the Stiefel manifold (where $A = -A^\dagger$). It is however a geodesic with respect to the canonical metric of the Stiefel manifold, and can certainly be used as viable retraction also in combination with the Euclidean metric.[1] The retraction in Eq. (9) requires the matrix exponential of $Q_X$, which can be evaluated with $O(np^2 + p^3)$ operations (compared to a naive $O(n^3)$ implementation) by exploiting the fact that the maximal rank of $Q_X$ is $2p$.[2] Another notable option for retraction is to replace the exponential in Eq. (9) by a Cayley transform, which can then exploit the reduced rank via the Sherman–Morrison-Woodbury formula, see Ref. 33 for details. While the latter can be somewhat faster, we use the retraction in Eq. (9) throughout this manuscript.

The above definitions constitute the bare minimum to formulate a Riemannian gradient descent algorithm on a Stiefel or Grassmann manifold. To exploit information from previous optimization steps, as happens in the conjugate gradient and quasi-Newton algorithms, one more ingredient is needed: a vector transport to transport gradients and other tangent vectors from previous points on the manifold to the current point. A vector transport generalizes the concept of parallel transport, and needs to be compatible with the chosen retraction. If $V = R_W(X, \alpha)$ is the end point of a retraction, a vector transport maps a tangent vector $Y \in T_W$ at the initial point to a tangent vector $\mathcal{T}_W(Y, X, \alpha) \in T_V$. As with the retraction, many choices are possible, but we use the transport

$$\mathcal{T}_W(Y, X, \alpha) = e^{\alpha Q_X} Y, \tag{10}$$

where $Q_X$ is as in Eq. (9), both for the Stiefel and the Grassmann case. This choice can be implemented efficiently, again by exploiting the low-rank property of $Q_X$. It has the additional benefit that it is a metric connection, which is to say it preserves inner products between tangent vectors, i.e. $g_W(Y_1, Y_2) = g_V(\mathcal{T}_W(Y_1, X, \alpha), \mathcal{T}_W(Y_2, X, \alpha))$. This simplifies some steps of the optimization algorithms and guarantees desirable convergence properties [33]. Note that Eq. (10) is not the parallel transport with respect to the Euclidean metric $g$ (nor with respect to the canonical metric), as it corresponds to a metric connection which has torsion, but this does not hinder its usage in optimization algorithms. Alternatively, one could again replace the exponential in Eq. (10) by a Cayley transform, if this was also done in the retraction.

## 2.4 Product manifolds

Note, finally, that a function depending on several isometries or unitaries corresponds to a function on the product manifold $\mathrm{St}(n_1, p_1) \times \mathrm{St}(n_2, p_2) \times \ldots$ (with $\times$ being the Cartesian product), where some of the factors could also be Grassmann manifolds instead. The corresponding tangent space is the Cartesian product of the individual tangent spaces (which corresponds to the direct sum as long as the number of tensors remains finite) and all of the above structures and constructions extend trivially.

---

[1]A closed form expression for the geodesics of the Stiefel manifold with respect to the Euclidean metric is also known, but cannot be written using a unitary applied to $W$; we refer to Ref. 32 for further details.

[2]How this is done depends slightly on the manifold. In the simpler Grassmann case, the exponential in Eq. (9) reduces to sines and cosines of singular values of $X$, and we can avoid constructing $W_\perp$ explicitly [32]. In the Stiefel case, we need to extract $W_\perp$ from a QR decomposition of $\begin{bmatrix} W & Z \end{bmatrix}$, where $Z = (\mathbb{1} - WW^\dagger)X = W_\perp B$, and compute the matrix exponential of $\begin{bmatrix} A & -B^\dagger \\ B & 0 \end{bmatrix}$. The full details can be found in the source code of the TensorKitManifolds.jl [34] package.

# 3   Riemannian gradient optimization

Having established the Riemannian geometry of Grassmann and Stiefel manifolds (and products thereof) in the previous section, we can now discuss how to implement Riemannian versions of some well-known gradient-based optimization algorithms, all of which are described in the literature [32, 33, 35–38].

We aim to minimize a cost function $C(W)$ defined on our manifold, where we consider a single argument $W$ for notational simplicity. The simplest approach is the Riemannian formulation of gradient descent, often also referred to as steepest descent. It is an iterative procedure which at every step computes the gradient of $C$ at the current point on the manifold, and then uses the chosen retraction in the direction of the negative gradient to find the next point. In steepest descent, the step size $\alpha$ is chosen so as to minimize $C$ along the retraction $\alpha \mapsto R_W(X, \alpha)$ with $X = -G$.

Finding $\alpha$ is known as the linesearch, and various algorithms and strategies exist for it. It is often unnecessary or even prohibitive to determine the minimum accurately; rather an approximate step size $\alpha$ that satisfies the Wolfe conditions [39] is sufficient to guarantee convergence. If we define $W' = R_W(X, \alpha)$ to be the new isometry, $G'$ the gradient at $W'$, and $X' = dR_W(X, \alpha)/d\alpha$ the local tangent to the retraction, then the Wolfe conditions are

$$C(W') < C(W) - c_1 g_W(G, X), \tag{11}$$

$$g_{W'}(G', X') > c_2 g_W(G, X), \tag{12}$$

with $0 < c_1 < c_2 < 1$ being free parameters [37, 38]. Eq. (11) states that the cost function should decrease sufficiently, while Eq. (12) says that its slope (which starts out negative for a descent direction) should increase sufficiently. Throughout our simulations, we use the linesearch algorithm described in Refs. 40, 41, which also takes into account that the descent property of Eq. (11) (also known as the Armijo rule) cannot be evaluated accurately close to convergence due to finite machine precision, and switches to an approximate but numerically more stable condition when necessary. In practice, a small number (often two or three) function evaluations suffice to determine a suitable step size $\alpha$.

While (Riemannian) gradient descent with step sizes that satisfy the Wolfe conditions converges in theory, this convergence is only linear and can be prohibitively slow, especially for systems of physical interest exhibiting strong correlations (e.g. critical systems) [3]. An improved algorithm with nearly the same cost is the nonlinear conjugate gradient algorithm, which dates back to the work of Hestenes and Stiefel. In conjugate gradient the search direction is a linear combination of the (negative) gradient and the previous search direction, a concept known as "momentum" in the context of optimizers for machine learning. Various schemes exist for the choice of the $\beta$ coefficient in this linear combination, see Ref. 44 and references therein. All these schemes can be applied in the Riemannian case, although the inner products that need to be computed as part of $\beta$'s definition need to be replaced by the metric $g$. Furthermore, to build a linear combination between the current gradient and the previous search direction, one needs to invoke the vector transport $\mathcal{T}$ from Sec. 2 for the latter to represent a valid tangent vector at the new base point. In the simulations below, we use the conjugate gradient scheme of Hager and Zhang [40, 41].

From a second order expansion of the cost function around the current point, one arrives at Newton's method, which suggests taking a step of length 1 in the direction of $-H^{-1}(G)$, where $H$ is the Hessian, i.e. the matrix of second derivatives. While Newton's method has a theoretical quadratic convergence rate close to the minimum, computing $H$ and its inverse is

---

[3]This can be argued by noting that the Hessian of the corresponding energy function is often related to the dispersion relation of the physical excitations in the system [42, 43], and thus has (near)-zero modes for such systems.

often prohibitively expensive and has various other issues. The Hessian might not be positive definite far away from the minimum, and furthermore depends on the second order behaviour of the retraction when formulating a Riemannian generalization of Newton's method. Quasi-Newton methods, on the other hand, construct an approximation to $H^{-1}$ using only gradients, computed at the successive points $W_k$ along the optimization. The most commonly used is the Limited-memory Broyden–Fletcher–Goldfarb–Shanno (L-BFGS) algorithm [39, 45], which keeps a low-rank, positive semi-definite approximation of $H^{-1}$ in memory. The Riemannian formulation of it also depends on the vector transport and has been well established, see Refs. 37, 38 and references therein.

Both the conjugate gradient and L-BFGS algorithms converge to a local minimum at a rate that is somewhere between the linear convergence of gradient descent and quadratic convergence of Newton's method. Which one is to be preferred often depends on the application. The latter requires a few more vector operations, but can use these to scale the inverse Hessian so that step size $\alpha = 1$ is typically accepted and no linesearch is needed in most iterations.

Despite the speedup provided by the conjugate gradient and L-BFGS algorithms, it is often beneficial to apply a *preconditioner* to the optimisation. A preconditioner is a transformation that maps one tangent vector to another, $X \mapsto \tilde{X}$, and that is applied when choosing the search direction. Using a preconditioner with gradient descent simply means retracting in the direction of the negative *preconditioned* gradient $-\tilde{G}$, instead of $-G$. Using preconditioners with conjugate gradient and quasi-Newton methods is not much more complicated, and we direct the reader to the numerical optimisation literature [39, 46, 47] for the details.

The choice of the preconditioner $X \mapsto \tilde{X}$ is typically guided by trying to capture some structure of the Hessian. The inverse Hessian $\tilde{X} = H^{-1}(X)$ would often be an ideal preconditioner, and while it is usually infeasible to implement, using some approximation to it may already help convergence significantly. A preconditioner (assumed to be positive definite) can also be seen as changing the metric in the problem, hopefully in such a way that the optimisation landscape becomes less singular and hence easier to navigate for the chosen optimisation algorithm. This geometrical viewpoint is illustrated in Fig. 1, and is what we will use to justify the preconditioners we use in our tensor network optimisations. Note that the same effect could be achieved by actually defining a new metric on the relevant Stiefel or Grassmann manifold, and repeating the steps in Sec. 2 again for this metric. However, we find that using the Euclidean inner product with an additional explicit preconditioning step gives greater flexibility without complicating e.g. the metric condition for the vector transport.

In the context of tensor networks, the cost function $C$ will typically be $C(W) = \langle \psi(W) | H | \psi(W) \rangle$, where $H$ is a local Hamiltonian, and $|\psi(W)\rangle$ is a tensor network state dependent on the isometry $W$. A tangent vector $X \in T_W$ can then be related to a state $|\Phi_W(X)\rangle = X^i | \partial_i \psi(W)\rangle$ in Hilbert space, which yields an induced inner product $\langle \Phi_W(X) | \Phi_W(Y) \rangle$ between tangent vectors $X, Y \in T_W$. A suitable preconditioner can then be extracted from the explicit expression of $\langle \Phi_W(X) | \Phi_W(Y) \rangle$, or some approximation thereof. As discussed in the applications below, we assume that this inner product can be written as $\langle \Phi_W(X) | \Phi_W(Y) \rangle \approx \mathrm{Tr}[X^\dagger Y \rho_W]$ for some $W$-dependent, hermitian, positive (semi)-definite $\rho_W$ of size $p \times p$. We can then implement a preconditioning step $X \mapsto \tilde{X} \in T_W$ such that (henceforth omitting the $W$ dependence)

$$\mathrm{Re}\,\mathrm{Tr}[Y^\dagger \tilde{X} \rho] = \mathrm{Re}\,\mathrm{Tr}[Y^\dagger X] \quad \forall\, Y \in T_W\,. \tag{13}$$

In other words, the Euclidean inner product with $X$ equals the more physically motivated inner product with $\tilde{X}$. If we express $X$ as $X = WA + W_\perp B$, where $A$ is skew-hermitian (Stiefel) or zero (Grassmann), then the solution to Eq. (13) is

$$\tilde{X} = W\tilde{A} + W_\perp \tilde{B}\,, \tag{14}$$

$$\text{where } \tilde{A}\rho + \rho\tilde{A} = 2A \text{ and } \tilde{B} = B\rho^{-1}\,. \tag{15}$$

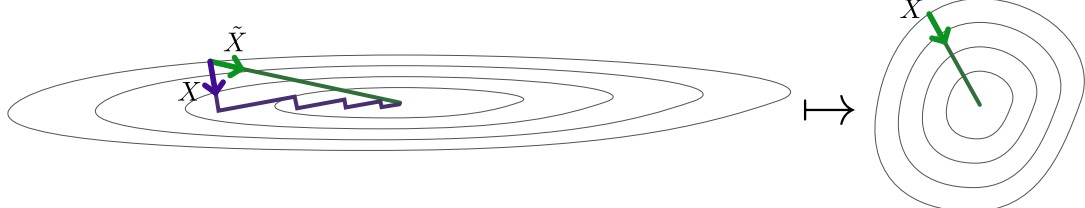

Figure 1: On the left, the grey ovals are the contour lines of the cost function in this 2-dimensional optimisation problem. The purple arrow $X$ is the negative gradient, and the zig-zag line emanating from it is the path that gradient descent takes. The relatively slow convergence of the gradient descent path is a consequence of the near-singular geometry of the contour lines, where the cost function varies much more along one axis than the other. The green arrow $\tilde{X}$ would be the optimal choice for the preconditioned search direction, the one that takes us to the optimum in a single retraction. By changing the geometry (i.e. the metric) of the problem, in this case by a simple rescaling of the axes, we can map to the problem on the right, where the geometry of the contour lines has become less singular. In this new geometry $\tilde{X}$ is in fact the negative gradient. This suggests that a preconditioner that implements this change of geometry would probably be beneficial for convergence. While the above is a cartoon example, redefining the metric to make the optimisation landscape less singular can be a useful way to design preconditioners more generally.

The equation for $\tilde{A}$ is a Sylvester equation, that can be solved easily and efficiently using e.g. an eigendecomposition of $\rho$ at a cost $O(p^3)$. The matrix $\rho$ may often be quite ill-conditioned, and in practice we have found the regularized inverse $\left(\rho^2 + \mathbb{1}\delta^2\right)^{-\frac{1}{2}}$ to work well. We discuss the choice of $\delta$ in the applications below.

Note that this preconditioner accounts for the structure of the physical state space, i.e. it corresponds to the induced metric of the variational manifold in Hilbert space. When implemented exactly, the preconditioned gradient corresponds to the direction in which a state would evolve under imaginary time evolution as implemented by the time-dependent variational principles of Dirac, Frenkel or McLachlan (see Refs. 22, 48 and references therein) and has been used with MPS as such [23]. This choice, or (block)-diagonal approximations thereof, as discussed in the next section for the case of MERA, was recently referred to as the "quantum natural gradient" in the context of variational quantum circuits [49]. This choice of preconditioner is independent of the Hamiltonian, and it is conceivable that a much bigger speedup can be obtained by explicitly taking the Hamiltonian into account. Such an improved preconditioner can probably not be implemented efficiently without resorting to an iterative linear solver, such as the linear conjugate gradient method. Such a scheme would be close in spirit to the set of optimization methods known as truncated Newton algorithms [46, 50]. The above preconditioner can then still prove useful to speed up this inner linear problem. We elaborate on this in the discussion in Section 6.

## 4 Application: MERA

In this section we show how Riemannian optimization methods can be applied to the multiscale entanglement renormalization ansatz (MERA), and demonstrate that the resulting algorithm outperforms the usual Evenbly-Vidal optimization method used for MERA. Specifically, we concentrate on a one-dimensional, infinite, scale invariant, ternary MERA, but the generalization to other types of MERAs is trivial.

A MERA is a tensor network of the form

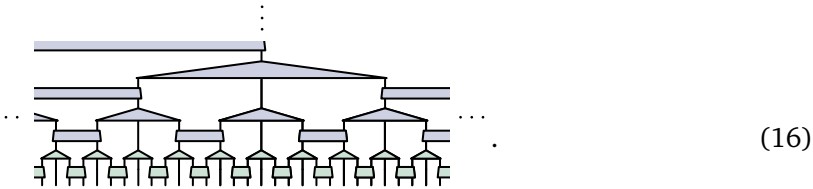

$$. \tag{16}$$

Each tensor in a MERA is isometric in the sense that

$$\text{and} \tag{17}$$

where red borders denote complex conjugation. The network defines a quantum state $|\text{MERA}\rangle$ living on the lattice at the bottom legs in Eq. (16). In the example MERA from Eq. (16), there are two distinct layers: There is one transition layer at the bottom, followed by a scale invariant layer, copies of which repeat upwards to infinity. Each layer $i$ is translation invariant and defined by two tensors, the disentangler $u_i = $ and the isometry $w_i = $. The cost function we are trying to minimize is $\langle \text{MERA}|H|\text{MERA}\rangle$, where $H = \sum_i h_i$ is a given local Hamiltonian. In our benchmark simulations we use the critical Ising Hamiltonian

$$h_i = -X_i X_{i+1} - Z_i . \tag{18}$$

The parameter space in which we are optimising is $\bigtimes_v M_v$, where $\bigtimes_v$ denotes Cartesian product over all the different tensors $v = u_1, w_1, u_2, w_2, \dots$, and $M_v$ is the Stiefel or Grassmann manifold of each tensor $v$. Any unitary one-site rotation on the top index of an isometry $w_i$ can be absorbed into the disentangler $u_{i+1}$ above it, and hence the natural manifold for $w$'s is the Grassmann manifold: $M_{w_i} = \text{Gr}$.[4] The same is not true for the disentanglers, for which similar unitary rotations would entangle the two top indices, and hence we treat them as points on Stiefel manifolds: $M_{u_i} = \text{St}$.[5] We have omitted the dimensions of the manifolds, since they depend on the physical site state space dimension $d$ and the bond dimension $D$ of the upper layers.

As discussed at the end of Section 2, the tangent space is the Cartesian product of the tangent spaces of the individual tensors, $\bigtimes_v T_v$, which corresponds to a direct sum structure, and the Riemannian geometry and associated operations extend trivially. The inner product, in particular, is the sum of the inner products on the individual manifolds.

To compute the gradients, we first discuss the partial derivatives. Hereto, we denote the partial derivative of the state $|\text{MERA}\rangle$ with respect to a tensor $v$ by $\partial_v |\text{MERA}\rangle$. Since each tensor appears several times in the network, $\partial_v |\text{MERA}\rangle$ has several terms in it, e.g.

$$\partial_{w_1} |\text{MERA}\rangle = \quad + \quad + \quad + \dots . \tag{19}$$

The partial derivative of the cost function is then $D_v = 2\partial_{v^\dagger} \langle \text{MERA}|H|\text{MERA}\rangle$. Up to a scalar factor, the same object arises in the context of the usual Evenbly-Vidal optimization algorithm, where it is called the "environment" of tensor $v$. These environments can be computed efficiently, and we refer the reader to Ref. 19 for how to do so. Extra care needs to be taken when dealing

---

[4]Note that we are not saying here that any member of the equivalence class $[w_i] = \{w_i U \mid UU^\dagger = U^\dagger U = \mathbb{1}\}$ leads to the same MERA: This is obviously not the case. Instead what we are saying is that changes of the form $w_i \mapsto w_i U$ are degenerate from the point of view of our optimization, as they can be cancelled by a corresponding change in one of the disentanglers. In other words, any tangent directions that correspond to changes of the type $w_i \mapsto w_i U$ are of no interest to us, and can be projected out.

[5]Indeed, $\text{Gr}(n,n)$ is the trivial singleton manifold $[\mathbb{1}]$.

with the scale invariant layer, something we discuss in Appendix B. The gradient $G_\nu$ is the projection of the partial derivative $D_\nu$ onto the tangent space $T_\nu$, as in Eq. (4) and (5). The total gradient $G$ of the whole parameter space is $G = (G_{u_1}, G_{w_1}, G_{u_2}, \dots) \in \times_\nu T_\nu$.

As mentioned above, the inner product between two tangents $X, Y \in \times_\nu T_\nu$ is $\sum_\nu g_\nu(X_\nu, Y_\nu)$, where $g$ is the Euclidean metric. However, each $X_\nu$ is associated with a state in the physical Hilbert space, schematically denoted as $\frac{\partial |\text{MERA}\rangle}{\partial \nu} X_\nu$, and we would like to implement a preconditioning that would equate to using instead a metric arising from the physical inner product, namely

$$\sum_{\nu,\nu' \in \{u_1, w_1, \dots\}} X_\nu^\dagger \frac{\partial^2 \langle \text{MERA}|\text{MERA}\rangle}{\partial \nu^\dagger \partial \nu'} Y_{\nu'}. \tag{20}$$

The cross-terms in this sum are quite expensive to compute, so we settle instead for the diagonal version

$$\sum_{\nu \in \{u_1, w_1, \dots\}} X_\nu^\dagger \frac{\partial^2 \langle \text{MERA}|\text{MERA}\rangle}{\partial \nu^\dagger \partial \nu} Y_\nu = \sum_{\nu \in \{u_1, w_1, \dots\}} \text{Tr}[X_\nu^\dagger Y_\nu \rho_\nu], \tag{21}$$

where $\rho_\nu$ is the reduced density matrix on the top index or indices of $\nu$. As discussed at the end of Sec. 3, preconditioning with this type of metric can be efficiently implemented for both Stiefel and Grassmann tangents. The regularization parameter $\delta$ used in computing the regularized inverse of $\rho$ (or the equivalent thereof for the Sylvester problem) in the preconditioner can also be allowed to vary. In particular, using a very small value of $\delta$ can be detrimental to the optimization in the beginning, when we are far from the minimum, and we have found $\delta = \|X_\nu\|$ to be a good choice.

To the best of our knowledge, the only algorithm that has systematically been used to minimize $\langle \text{MERA}|H|\text{MERA}\rangle$ is the Evenbly-Vidal algorithm, described in detail in Ref. 19. Fig. 2 shows benchmark results comparing the Evenbly-Vidal algorithm and an L-BFGS optimization, using the above preconditioning. The L-BFGS optimization converges significantly faster for all the simulations displayed in the figure. Note the logarithmic scale of the horizontal axis, which allows to visualize both the initial and final parts of the convergence. Individual iterations take somewhat longer to run with L-BFGS (though the asymptotic complexity remains the same, $O(D^8)$ for the ternary MERA), typically 1.5–2 times longer in our simulations, but this effect is more than compensated for by the faster rate of convergence [6].

While our benchmark is the Ising model with a ternary MERA, we find qualitatively similar results for binary MERAs, and for different models such as the XXZ model. Moreover, we show results for the L-BFGS algorithm as they are slightly better than those of the conjugate gradient method, but the difference is not drastic. Other small changes, such as treating the isometries as elements of Stiefel manifolds, or using different retractions or the canonical metric, have limited effects on the results. The use of preconditioning with the Hilbert space inner product, however, is crucial, and thus indicative that further improvements could be made by improving the preconditioner. Note that MERA optimizations are somewhat prone to getting stuck in local minima, especially at higher bond dimensions, something that affects all optimization methods we have tried.

The strategy of the Evenbly-Vidal algorithm is similar to alternating least-squares algorithms: At every step a single tensor of the network is updated, while considering the other tensors as independent of it. The specific update needs to account for the isometry condition and is

---

[6]Note that the speed difference between the Evenbly-Vidal algorithm and gradient methods depends somewhat on the type of MERA. The most costly operations inherent to the gradient methods (retractions, vector transport and applying the preconditioner) scale as $O(D^6)$, whereas the leading-order cost of both algorithms (computing energy and gradients or environments) is $O(D^7)$ for modified binary, $O(D^8)$ for ternary, and $O(D^9)$ for binary MERA. The higher scaling of e.g. binary versus ternary MERA is compensated for by ternary MERAs typically needing higher bond dimensions to achieve the same accuracy, which shows as proportionally higher subleading costs.

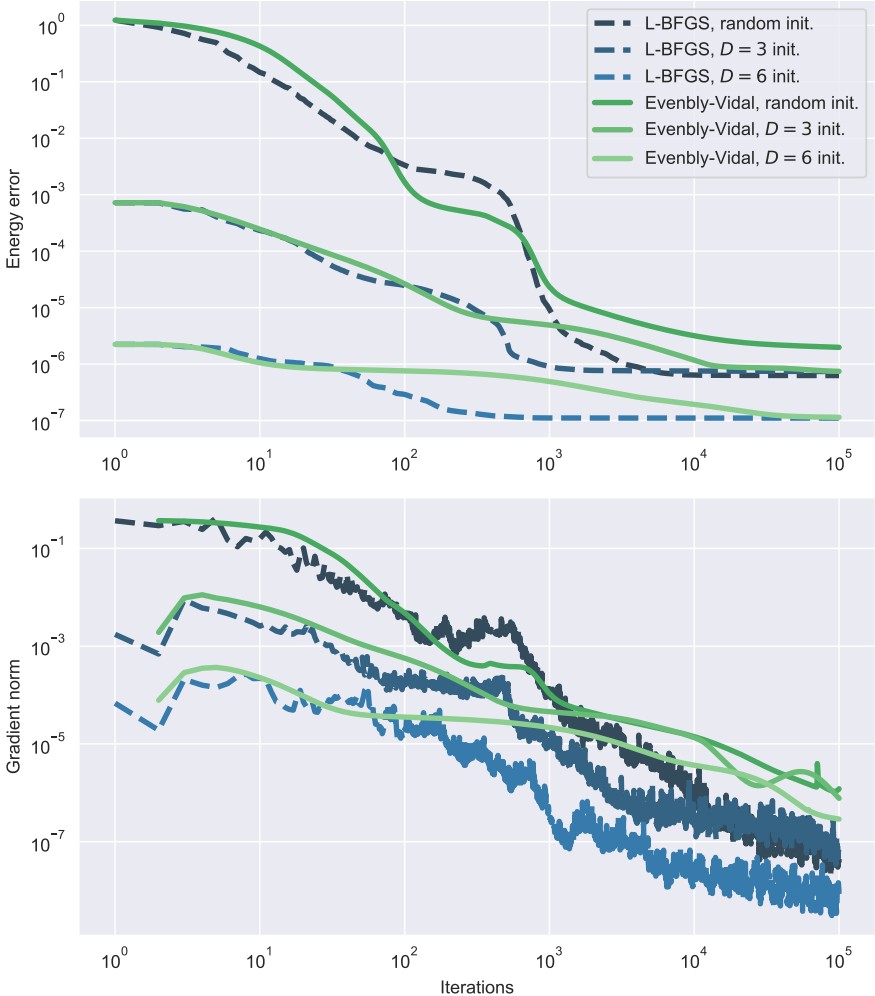

Figure 2: A comparison of convergence in optimising a MERA using the Evenbly-Vidal algorithm (solid green lines) and L-BFGS on Riemannian manifolds (dashed blue lines). Displayed here are the ground state energy error compared to the exact value (top) and the norm of the gradient (bottom). The benchmark model in question is the critical Ising model. In all simulations the MERA is a bond dimension 8 ternary MERA with two transition layers, with the $\mathbb{Z}_2$ symmetry enforced. For both algorithms three different simulations are shown, corresponding to three different starting points: One was a MERA initialized with random isometries and identity disentanglers, the two others were MERAs optimized to convergence at a lower bond dimension, $D = 3$ and $D = 6$, and then expanded to the full bond dimension $D = 8$. This kind of slow ramping up of the bond dimension can be useful for both speed of convergence and for avoiding local minima. As the energy error plot shows, here, too, some simulations converge to a local minimum instead of the global one. In all cases the convergence speed of L-BFGS algorithm clearly outperforms the Evenbly-Vidal algorithm.

reviewed in Appendix A. An update like this typically brings down the energy at every step, and the procedure is then iterated over all the different tensors until convergence. At first this seems entirely different from gradient optimization: The Evenbly-Vidal algorithm makes discontinuous jumps from one point in the parameter space to another, one tensor at a time, whereas gradient methods perform smooth retractions of all the tensors at once. However, hidden in the Evenbly-Vidal update is in fact a kind of step size parameter, that is the additive scale of the effective Hamiltonian. In Appendix A we show that there is a particular limit in which the Evenbly-Vidal algorithm reduces to gradient descent preconditioned with the metric from Eq. (21). Although this limit is not necessarily where the algorithm is typically run, this relation to a first-order optimisation method gives some intuition for how a quasi-Newton or conjugate gradient method could outperform it.

## 5 Application: MPS

In this section we show how gradient optimization methods on Riemannian manifolds can be applied to optimize a matrix product state (MPS). The MPS is kept in its left-canonical form, where each tensor is an isometry from its physical index and left virtual index to its right virtual index. Such an MPS can be depicted as

$$\cdots \rightarrow \triangleright \triangleright \triangleright \cdots \, , \tag{22}$$

where

$$ = \, , \tag{23}$$

and red borders denote complex conjugation. Every injective MPS can be gauge-transformed into this form. For simplicity's sake we concentrate on the case of an infinite MPS with one-site translation symmetry [28, 51]. Such an MPS is defined by a single isometry. However, the generalization to a finite MPS or to one with a larger unit cell is straightforward.

We consider the tensor $\triangleright$ defining the MPS as a point on a Grassmann manifold, since unitary rotations on the right virtual index of each tensor are mere gauge transformations, which can be absorbed in the next tensor without changing the physical state. The inner product between two tangent tensors, as well as retraction and transport functions are as explained in Sec. 2, but see also Ref. 52 for further details about the Riemannian geometry of MPS manifolds. The cost function is the expectation value of a Hamiltonian, which we represent as a matrix product operator (MPO)

$$\cdots \rightarrow \bigcirc \bigcirc \bigcirc \cdots \, . \tag{24}$$

The partial derivative of the cost function with respect the isometry can be computed as

$$2 \cdot \left( H_l \bigcirc H_r \right) \, , \tag{25}$$

where $H_l$ and $H_r$ are the left and right energy environments, which can be efficiently be computed as outlined in Refs. 2, 28, 51. The partial derivative can then be projected onto the tangent space of the Grassmann manifold, as in Eq. (5), to obtain the gradient.

For preconditioning, we want the effective inner product between two tangent vectors for an individual site, $\text{▷}$ and $\text{▷}$, to be

$$\sum_{n=-\infty}^{\infty} \cdots \overset{n}{\cdots} \cdots = \cdots \cdots = \cdots . \tag{26}$$

Here $n$ is the separation between the sites, and the first equation follows from the fact that Grassmann tangent vectors are orthogonal to the Grassmann-points they are at, i.e. Eq. (3). This is known as the left gauge condition for tangent vectors in the context of MPS [51, 52]. The tensor at the very right in Eq. (26) is the dominant right eigenvector of the MPS transfer matrix,

$$\text{▷}\diamond = \diamond\, , \tag{27}$$

and plays the role of $\rho$ from Eq. (13). In contrast to the MERA case, this expression corresponds to the exact Hilbert space inner product between tangent vectors, without approximations. Implementing preconditioning with this inner product requires only implementing the map

$$\text{▷} \mapsto \text{▷}(\diamond)^{-1}. \tag{28}$$

As with MERA, regularising the inverse of the right eigenvector is paramount for performance, especially during the initial iterations of the optimization process. In the MPS case we use the regularisation $\left( \diamond + \mathbb{1}\delta \right)^{-1}$ with $\delta = \left\| \text{▷} \right\|^{2}$.

We would like to note that, with an exact inverse (i.e. $\delta = 0$) in Eq. (28), standard gradient descent in the limit of a small step size $\alpha \to 0$ amounts to imaginary time evolution, implemented using the TDVP [23]. This is a consequence of the Kähler structure of the MPS manifold [22, 51, 52].

With the above building blocks, we are ready to use Riemannian gradient methods for a uniform MPS. For benchmarking, we compare against the well-established VUMPS algorithm [28]. We are *not* able to consistently outperform VUMPS for all MPS problems, but we are able to do so for some problems. As an example of a case where gradient optimization performs well, we consider the triangular lattice antiferromagnetic spin-$\frac{1}{2}$ Heisenberg model on a cylinder. The classical analogue of this model is disordered, but quantum fluctuations restore the order again in the infinite 2d plane. It is an example of order from disorder and has been studied extensively [53–57]. Considering the cylinder as a 1D system with longer range couplings ("coiling" around the cylinder), the Hamiltonian can be written as

$$H = \sum_{i}(h_{i,i+1} + h_{i,i+c} + h_{i,i+c+1}), \qquad h_{i,j} = X_i X_j + Y_i Y_j + Z_i Z_j\,, \tag{29}$$

where $X$, $Y$, and $Z$ are the spin operators. Here $c$ is the width of the cylinder, which we fix to $c = 6$ for our benchmark. The appropriate MPS ansatz for this model is a uniform MPS with $c$-site unit cell. We also enforce the SU(2) symmetry of the MPS, since continuous symmetry breaking does not take place for finite $c$.

In Fig. 3 we show results comparing VUMPS with both gradient descent and conjugate gradient optimizations, with the above preconditioner. VUMPS does clearly better in the beginning of the optimization, which starts from a randomly initialized MPS. However, its convergence speed after the initial burst is similar to that of gradient descent, whereas conjugate gradient converges at a clearly faster rate. This is to be expected, as VUMPS was inspired by

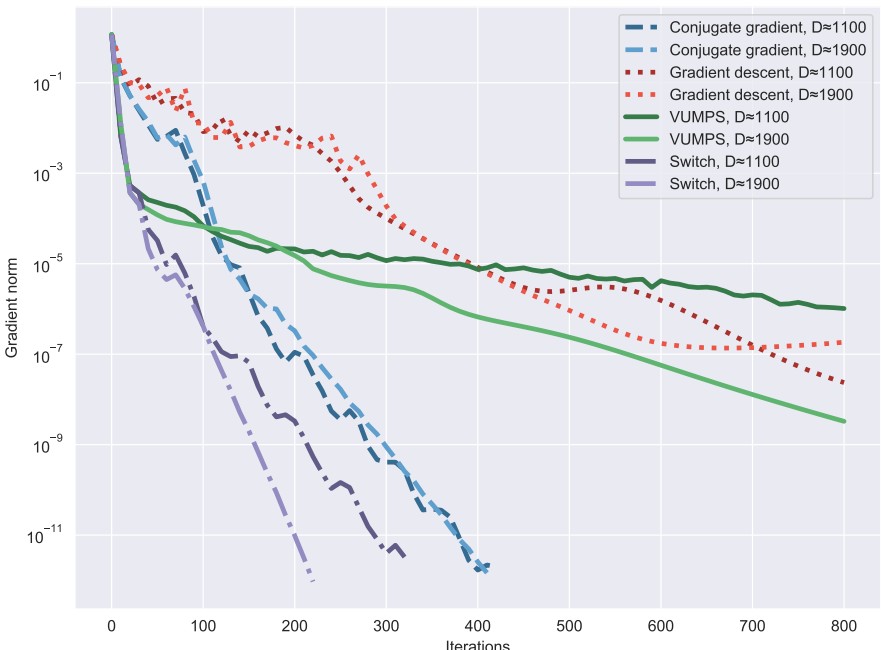

Figure 3: A comparison of convergence in optimising an infinite MPS using VUMPS (solid green lines), conjugate gradient (dashed blue lines), gradient descent (dotted red lines), and the "switch" method that combines VUMPS and conjugate gradient (dash-dotted purple lines). The benchmark model in question is a triangular lattice antiferromagnetic spin-$\frac{1}{2}$ Heisenberg model on a cylinder of width 6. Results are shown for MPS bond dimensions 1100 (darker colors) and 1900 (lighter colors). SU(2) symmetry of the tensors is enforced. VUMPS clearly performs the best at the start of the optimization, but its asymptotic convergence rate is roughly the same as that of gradient descent, whereas conjugate gradient can be seen to converge significantly faster. A best-of-both-worlds solution is the switch method, which does 30 iterations of VUMPS at the start and then switches over to conjugate gradient. L-BFGS produces results roughly comparable to those of conjugate gradient, but we do not show them here.

imaginary time evolution using the TDVP (or thus, Riemannian gradient descent), and should become equivalent to it for small step sizes, i.e. when the algorithm is close to convergence. Note that convergence in Fig. 3 is shown with respect to number of iterations, not actual running time. VUMPS iterations, which internally use an iterative eigenvalue solver, take roughly 1.5 times as long as conjugate gradient iterations, thus increasing the gap between the two methods when plotting with respect to running time. Finally, we have also included results for a method labeled "switch", where we use VUMPS for the first few iterations, and then switch over to conjugate gradient, which outperforms both of the individual methods.

As mentioned, the Riemannian optimization methods explained here can be easily applied to a finite MPS as well. Preliminary benchmarks indicate that for some models, gradient methods can outperform the DMRG algorithm [18]. The qualitative picture is similar to what we observe with infinite MPS, where variational methods like VUMPS and DMRG are superbly fast at making progress early in the optimization, but if the problem is difficult and a slow convergence sets in, the asymptotic convergence rate of preconditioned conjugate gradient or L-BFGS is often better. We leave, however, a more detailed study of finite MPS optimization for future work.

# 6   Conclusion

The MERA and MPS results of Secs. 4 and 5 illustrate that Riemannian gradient-based optimization can be a competitive method for optimising tensor network ansätze. Partial derivatives of the energy with respect to a given tensor give rise to tensor network diagrams that also appear in current algorithms such as the Evenbly-Vidal algorithm for MERA and the VUMPS algorithm for infinite MPS. Implementing these methods is thus only a matter of computing an actual update direction from the computed gradient using the recipe of the chosen method (gradient descent, conjugate gradient or L-BFGS quasi-Newton), and replacing the update step with a retraction. Vector transport is subsequently used to bring data from the previous iteration(s), such as former gradients, to the tangent spaces at the current iterate.

This approach is fully compatible with exploiting the sparse structure of tensors arising from symmetries, such as $\mathbb{Z}_2$, $\mathsf{U}_1$ or even non-abelian symmetries such as $\mathsf{SU}_2$. The isometry condition defines how the tensor should be interpreted as a linear map, so that, when using symmetric tensors, they take a block diagonal form in a basis of fused representations, according to Schur's lemma. The isometry condition itself, the projection onto the tangent space, the retraction, and the vector transport then all apply at the level of those individual diagonal blocks, and can easily be implemented as such. Indeed, as mentioned, $\mathbb{Z}_2$ symmetry was used in the MERA results and $\mathsf{SU}_2$ symmetry in the MPS results presented above.

We have demonstrated the usefulness of gradient optimization for MPS and MERA, but there are other tensor network methods that also involve isometric tensors. Notable cases we have not discussed are tree tensor networks, i.e. MERA without the disentanglers, and the tensor network renormalization algorithm [5] (TNR), which is closely related to MERA, and for which the usual optimization method is a variant of the Evenbly-Vidal algorithm. We expect that in both these cases gradient methods could provide similar advantages as they do for MERA.

While we focused here on the application of Riemannian gradient-based optimization methods for tensor networks with isometry constraints, even their Euclidean counterparts have not received a great deal of attention as an alternative to the standard recipe of optimizing individual tensors in an alternating sequence using only local information (i.e. from the current iteration, not relying on the history of previous iterations). While the latter can be expected to work extremely well when correlations are relatively short-ranged, there is no particular reason that gradient-based methods which optimize all tensors simultaneously could not replicate this behaviour in this regime, when provided with a suitable preconditioner. However, gradient-based methods, in particular those that use a history of previous iterations, such as conjugate gradient and quasi-Newton algorithms, have the potential to also work in the regime with long-range and critical correlations. These conditions typically imply very small eigenvalues in the Hessian, which is detrimental for methods that only use first order information of the current iterate. A specific example includes situations of low particle density, for which specific multigrid algorithms have been explored [58]. It would be interesting to see if gradient-based algorithms would alleviate the problems that plague DMRG in this regime. Related to this is the case of continuous MPS [59], where the state is not even a linear or homogeneous function of the matrices containing the variational parameters and DMRG- or VUMPS-like algorithms are unavailable. In those cases, gradient-based methods are the only alternative [60,61].

For all of these applications, a well-considered preconditioner is of paramount importance. A suitably preconditioned gradient descent can easily outperform a conjugate gradient or quasi-Newton algorithm with ill-chosen parameterization. In the case of MPS-specific methods such as DMRG or VUMPS, this is implicit in using what is known as the center-gauge. For gradient methods, the same effect is accomplished by using the reduced density matrix which appears in the physical inner product of these tangent vectors in Hilbert space. However, it is conceivable

that there is plenty of room for improvement by using information of the actual Hamiltonian in constructing a preconditioner, i.e. by using its matrix elements with respect to the tangent vectors rather than those of the identity operator. While the full Hessian needed for Newton's algorithm can be computed for the case of MPS [43], this comes with a large cost and would likely be inefficient. A single application of the Hessian to a given tangent vector requires to solve several non-hermitian linear problems with iterative solvers (e.g. the generalized minimal residual algorithm), in order to obtain cubic scaling in the bond dimension. Hence, Newton's method would amount to three nested levels of iterative algorithms. A local positive definite approximation of the Hessian which can be applied to a given vector efficiently and directly can be constructed, by (i) ignoring contributions from taking both partial derivatives in the ket or in the bra (somewhat similar to the Gauss-Newton or Levenberg–Marquardt algorithms), as well as (ii) discarding non-local contributions similar to how we ignored off-diagonal contributions in the inner product of MERA tangent vectors. Such a preconditioner would still need an iterative solver (e.g. linear conjugate gradient) to be applied efficiently, but the improvement over the metric or preconditoner constructed here might be sufficiently significant to overcome this overhead. Indeed, such a scheme is similar in spirit to truncated Newton algorithms [50], for which dedicated implementations of the inner conjugate gradient method exist, which detect the absence of positive definiteness and produce valid descent directions at every step. A related strategy might be to directly use the solution of the local problem from DMRG, VUMPS or the Evenbly-Vidal algorithm as some kind of nonlinear preconditioner, as outlined in Ref. 47. These ideas will be explored in a forthcoming paper.

As a final remark, we would like to point out that the techniques explored in this manuscript are relevant beyond the case of tensor network representations of ground states of many body systems. Various tasks in quantum computation also rely on the classical optimization of the gates in a unitary circuit as a precursory step, and this particular classical task can likewise benefit from the Riemannian optimization methods on which we have reported.

*Note:* Near completion of this work, the preprint "Riemannian optimization and automatic differentiation for complex quantum architectures" by Luchnikov, Krechetov, and Filippov [62] appeared on the arXiv, which also proposes the use Riemannian optimization techniques for applications involving isometric tensor networks, quantum control and state tomography. In particular, they also consider Stiefel manifolds to perform gradient optimization on a (finite) MERA, although with different gradient-based algorithms inspired by machine learning. They do not consider the use of preconditioners nor applications to MPS, so that the two articles complement each other and pave the way for a bright future for Riemannian gradient-based optimization of tensor networks.

# Acknowledgements

We thank Glen Evenbly, Andrew Hallam, Laurens Vanderstraeten, and Frank Verstraete for useful discussions. We also thank Miles Stoudenmire and an anonymous referee for helpful feedback.

**Funding information** This work has received funding from the European Research Council (ERC) under the European Unions Horizon 2020 research and innovation programme (grant agreements No 715861 (ERQUAF) and 647905 (QUTE)), and from Research Foundation Flanders (FWO) via grant GOE1520N and via a postdoctoral fellowship of MH.

# A    Evenbly-Vidal algorithm and its relation to gradient descent

This appendix summarizes the local update in the Evenbly-Vidal algorithm, illustrates the implicit notion of a step size it contains, and relates it to a preconditioned gradient descent in the limit of small step size.

For a Hamiltonian $H$ the MERA cost function is

$$C(W) = \langle \text{MERA}(W)|H|\text{MERA}(W)\rangle, \tag{30}$$

where we have chosen to focus on a single isometry or disentangler $W$ only. Note that no normalization is necessary, as the state is properly normalized due to the isometry conditions on the tensors. Because $C$ is a homogeneous function of $W$, $C(W) \propto \text{Tr}[W^\dagger D]$, where $D = 2\partial_{W^*}C$ is the partial derivative that we used in the gradient optimization as well, also called the *environment* of $W$. Given this linear approximation of the cost function, where we assume $D$ to be independent of $W$ (which it in reality is not), the choice of $W$ that extremises it is $W = \pm Q$, where $D = QP$ is the polar decomposition, or as the original paper [19] expresses this, $Q = UV^\dagger$ where $D = USV^\dagger$ is the singular value decomposition. While the sign of $W$ matters for the linearized cost function, it does not for $C$, as $C$ contains only even powers of $W$. Although the assumption of $D$ being independent of $W$ is clearly false, the update that sets $W = Q$ still works as an iterative step, that in most situations increases $\|C\|$. This step can then be repeated, and performed in turn for each of the different tensors that make up the MERA, to converge to a local maximum of $\|C\|$. This algorithm has fixed points where $D = WP$, i.e. when $W$ equals the polar factor of $D$. In that case, it can easily be verified that the gradient $G$ associated to $D$ by orthogonal projection onto the Stiefel tangent space vanishes, which confirms the necessary condition that this scheme converges to local extrema.

However, in order to ensure that maximizing $\|C\|$ amounts to minimizing $C$, the Hamiltonian is redefined as $H_\gamma = H - \gamma\mathbb{1}$, with $\gamma$ sufficiently large, e.g. so as to make $H_\gamma$ negative definite. In that case, the ground state approximation is indeed the state that maximizes $\|C\|$.

Although $\gamma$ was introduced to shift $H$ by a constant to make it sufficiently negative, it turns out to play the role of an inverse step size. To see this, first note that

$$C_\gamma = \langle \text{MERA}(W)|H_\gamma|\text{MERA}(W)\rangle = C - \gamma\,\text{Tr}[W^\dagger W\rho], \tag{31}$$

where $\rho$ is the reduced density matrix at the top index or indices of $W$. Consequently, $D_\gamma = D - \gamma W\rho$. Now decompose $D$ as $D = W(A + S) + W_\perp B$, where $A$ and $S$ are the skew-hermitian and hermitian parts of $W^\dagger D$, and thus

$$D_\gamma = W(A + S - \gamma\rho) + W_\perp B = W(S - \gamma\rho) + G. \tag{32}$$

Here $G = WA + W_\perp B$ is the gradient, obtained by projecting $D$ onto the Stiefel tangent space at base point $W$. As expected, the term in the Hamiltonian $H_\gamma$ that is proportional to the identity operator does not contribute to the Stiefel gradient. At convergence, $A$ and $B$ will be zero and the role of $\gamma$ is clearly to shift the eigenvalues of $S$ so as to have a fixed sign.

Now consider a small but non-zero $G$, i.e. when the algorithm is close to convergence, and treat it as a perturbation to $W(S - \gamma\rho)$. To see how the Evenbly-Vidal update behaves in this case, we need to understand perturbation theory of the polar decomposition. If $X = QP$ is the polar decomposition of some arbitrary matrix $X$, and we perturb it as $X + \mathrm{d}X$, then an exercise that we omit here shows that

$$X + \mathrm{d}X = (Q + \mathrm{d}Q)(P + \mathrm{d}P), \tag{33}$$

where $dP$ is some hermitian matrix we do not care about, and

$$dQ = QA_X + Q_\perp B_X \tag{34}$$

$$\text{where} \quad A_X P + P A_X = Q^\dagger dX - dX^\dagger Q \tag{35}$$

$$\text{and} \quad B_X = Q_\perp^\dagger dX P^{-1}. \tag{36}$$

Matching this up with our case,

$$D_\gamma = \underbrace{W}_{=Q} \underbrace{(S - \gamma\rho)}_{=P} + \underbrace{WA + W_\perp B}_{=dX}, \tag{37}$$

we obtain

$$dQ = dW = WA_X + W_\perp B_X \tag{38}$$

$$\text{where} \quad A_X(S - \gamma\rho) + (S - \gamma\rho)A_X = 2A \tag{39}$$

$$\text{and} \quad B_X = B(S - \gamma\rho)^{-1}. \tag{40}$$

If we assume that $\gamma$ is sufficiently large so that $S$ is negligible compared to it, this becomes

$$dW = -\frac{1}{\gamma}(W\tilde{A}_X + W_\perp \tilde{B}_X) \tag{41}$$

$$\text{where} \quad \tilde{A}_X \rho + \rho \tilde{A}_X = 2A \tag{42}$$

$$\text{and} \quad \tilde{B}_X = B\rho^{-1}. \tag{43}$$

Comparing this with Eqs. (14) and (21), we can identify this with $dW = -\frac{1}{\gamma}\tilde{G}$, where $\gamma^{-1}$ thus plays the role of a step size in the Evenbly-Vidal algorithm, and $\tilde{G}$ is the gradient preconditioned with the same metric that was used in our gradient optimization in Sec. 4. Indeed, this observation further motivates our specific choice of preconditioner.

Note that in practice, the Evenbly-Vidal algorithm might not satisfy the assumption of large $\gamma$. The analysis above remains valid up to the final assumption, and might thus give an indication of a better preconditioner for MERA optimization that includes information from the Hamiltonian, yet can still be implemented efficiently. Instead of $\rho$, we could use $\rho - \gamma^{-1}S$, with $S$ the symmetric part of the $W^\dagger D$ and $\gamma$ chosen sufficiently big to ensure positive definiteness. We leave this proposal for future work.

# B  Efficient computation with the scale invariant layer of a MERA

In the optimization of an infinite MERA, the scale invariant layers at the top need to be treated somewhat differently from the rest. To discuss this, we first need to lay down some notation. We denote the local Hamiltonian term ascended to the lowest scale invariant layer by $h$. We often think of $h$ not as an operator $V \to V$, but as a vector in $V \otimes \bar{V}$, and denote this vector $\langle h|$. Similarly, we denote the local scale invariant density matrix $\rho$, and its vectorized version by $|\rho\rangle$. Finally, we call $A$ the ascending superoperator, thought of as a linear operator $V \otimes \bar{V} \to V \otimes \bar{V}$. Right-multiplying a vector like $\langle h|$ by $A$ corresponds to raising it by a layer, and left-multiplying a vector like $|\rho\rangle$ by $A$ corresponds to lowering it by a layer.

There are two problems that need to be solved for $A$ at every iteration of the optimization. First, to find $|\rho\rangle$, we must solve the eigenvalue equation $A|\rho\rangle = |\rho\rangle$. Second, when computing the gradient, we need to take the partial derivative $\partial_v \text{Tr}[h\rho] = \partial_v \langle h|\rho\rangle$, where $v$ is either the

disentangler or the isometry of the scale invariant layer. Expanding the dependence of $|\rho\rangle$, through $A$, on $v$, one finds

$$\partial_v \langle h|\rho\rangle = \sum_{i=0}^{\infty} \langle h|A^i (\partial_v A)|\rho\rangle. \tag{44}$$

To evaluate this we need to find the value of the series $\sum_{i=0}^{\infty}\langle h|A^i$. At face-value this diverges if $\langle h|$ has overlap with $\langle \mathbb{1}|$ (the vectorized version of the identity matrix), since $\langle \mathbb{1}|A = \langle \mathbb{1}|$. However, it turns out that any contributions to the partial derivative that are of the form $\langle \mathbb{1}|(\partial_v A)|\rho\rangle$ are orthogonal to the Grassmann/Stiefel tangent plane and thus projected out, because they correspond to shifting the cost function by a constant. Hence we can define $A' = A - |\rho\rangle\langle \mathbb{1}|$ and replace the above series by $\sum_{i=0}^{\infty}\langle h|A'^i$, which converges like a geometric series, since all eigenvalues of $A'$ are smaller than 1 in modulus. Indeed, this can similarly be understood as regular perturbation theory for the eigenvector $\rho$ of the (non-hermitian) operator $A$, whose eigenvalue 1 does not change under the perturbation.

All of the above is well-known from the original MERA papers [3,19], and comes down to solving two relatively simple linear algebra problems. The reason this is worth mentioning, is that multiplication by $A$ is the leading order cost of the whole MERA optimization, and thus as few such operations should be done as possible. With the traditional Evenbly-Vidal optimization, approximations have often been used, such as approximating $\langle h|$ at iteration $i$ as $\langle h_i| = \langle h_{i-1}| + \langle h_{i-1}|A'$, to save computation time [63]. With gradient algorithms like the ones presented here, these kinds of approximations may not be feasible, since the gradient needs to be computed to good accuracy at every step to be able to perform a line search. We have found, however, that using Krylov subspace methods for the eigenvalue problem $A|\rho\rangle = |\rho\rangle$ and the linear problem of solving $\langle h|\sum_{i=0}^{\infty}A'^i$ from $(\langle h|\sum_{i=0}^{\infty}A'^i)(\mathbb{1}-A) = \langle h|$, with a small Krylov space dimension (e.g. 4) and the solution from the previous iteration as the initial guess, leads to accurate results usually with very few applications of $A$. This helps make the MERA gradient optimization methods competitive with the Evenbly-Vidal algorithm.

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
