# Peer review of "Riemannian optimization of isometric tensor networks"

_SciPost Physics, doi:SciPost Phys. 10, 040 (2021)_

## Round 3 · Referee Report · Edwin Miles Stoudenmire (Referee 1) · 2020-12-17

Strengths

1 - the algorithms presented perform very favorably, often outperforming existing algorithms for the same problems
2 - the problems considered are central to the use of tensor networks in physics
3 - the paper is authoritative in its use and discussion of advanced mathematical concepts
4 - the authors provide demonstration software

Weaknesses

1 - because some of the content is highly technical, the paper could benefit from some more explanatory text in certain places, but the changes needed are small

Report

This is a well written paper which adapts and extends state-of-the-art optimization techniques developed in the applied math community to the setting of tensor networks. The authors review and adapt the theory of Riemannian optimization algorithms for manifolds of tensors that obey an isometric (one-sided unitary) constraint (Steifel manifold) or belong to an equivalence class of such tensors up to a unitary transformation (Grassmann manifold). Along with reviewing the relevant facts from the math literature, the authors discuss how to adapt these algorithms to the tensor network setting, such as how preconditioning maps can be defined appropriately. They also discuss the current state-of-the-art algorithm for these problems (the "Evenbly-Vidal algorithm") not only as a baseline, but also going into a theory of how this algorithm can be related to gradient optimization under certain conditions. The authors proceed to show two specific applications: optimizing a ternary MERA and an infinite MPS to be ground states of Ising model Hamiltonians.

The algorithms developed here perform very well. For MERA optimization, the present algorithms significantly outperform the Evenbly-Vidal algorithm in the later stages of optimization, requiring orders of magnitude fewer steps when starting from a random initialization. However, the Evenbly-Vidal algorithm remains competitive for the earlier stages of the optimization.

For MPS optimization, it is interesting to see that these more sophisticated gradient techiques can outperform VUMPS (a kind of improved infinite DMRG algorithm), at least when going for higher accuracies which are definitely a relevant case. I appreciate how the authors choose a more challenging model than the square-lattice transverse-field Ising model here.

Based on the important advances reported here, and the good quality of the paper, I recommend it to be accepted with minor revisions (see Requested Changes below). I believe it will be cited often and represents an important advance in the practice of using tensor networks.

Requested changes

Warnings issued while processing user-supplied markup:

  • Inconsistency: plain/Markdown and reStructuredText syntaxes are mixed. Markdown will be used.
    Add "#coerce:reST" or "#coerce:plain" as the first line of your text to force reStructuredText or no markup.
    You may also contact the helpdesk if the formatting is incorrect and you are unable to edit your text.

1 - Around Eqs (2) and (3), the matrix B is introduced to parameterize tangent vectors X. Presumably B is an arbitrary complex matrix, but this isn't explicitly stated. It would be a helpful clarification to say what kind of matrix B is allowed to be.

2 - An optional suggestion for the text just before and including Eqs (4) and (5) is to use a different letter than "D" for the complex matrix whose projection is being computed. Perhaps it could just be "M" for matrix? I understand that the matrix the authors have in mind to project is in fact a derivative as made clear in the next section, but I think the statements about (4) and (5) apply more generally to any matrix correct? So using a different letter would clarify the sense of generality to the reader.

3 - Just after Eq. (9) I found myself wondering how $W_\perp$ is determined in practice by the authors. I can think of various ways to do it, and I wonder if it could be helpful to a reader to know which one the authors selected, and if there is any reason for their choice.

4 - It would be helpful to have a sentence or two after Eq. (9) explaining how the authors compute the exponential of $Q_X$ in a way that takes advantage of its low rank. Though I can think of at least one way, I am not sure it is the one used by the authors, and so more details should (briefly) be given here or a reference cited.

5 - [This is the most important change I suggest:] At the end of Section 3, the authors presume that the reader is familiar with the idea of a gradient preconditioner, but I do not think that will typically be the case for the intended audience. Without looking up other literature on preconditioned gradient methods, the text does not give enough information about where and when the preconditioning map is applied during the optimization algorithm. While I do not think it is necessary for the authors to review the theory of preconditioning in a complete way, it would be very helpful here if they said a few more words about when the preconditioning map is applied. It could be explained, for example, in the context of plain gradient descent and then words said about how there are standard, analogous steps for conjugate gradient and L-BFGS.

6 - There seems to be an issue with the arxiv link for Ref. 32 (Edelman) (the URL should be https://arxiv.org/abs/physics/9806030)

Regarding the demo software

7 - there were a number of warnings printed when just doing "make" in the MPS folder on a Linux system. The warnings were related to precompilation. However it did work in spite of that.

8 - perhaps the authors could just provide Julia code with information about the depedencies? And similar for the Python code? This may be the case already, so could just involve putting instructions in the README files about how to run each code using the julia or python commands alone.

9 - I was able to successfully run the MPS/benchmark.jl code just using Julia on my Mac system, and it provided useful on-screen output that indicated the method was working and gave me a feel for its performance. However, the files in the logs/ folder afterward was empty. I wanted to mention this in case it is a bug or in case there could be a better default that would leave data in the log files so that it could be plotted.

  • validity: top
  • significance: high
  • originality: high
  • clarity: high
  • formatting: perfect
  • grammar: perfect

Author:  Markus Hauru  on 2020-12-23  [id 1106]

(in reply to Report 1 by Edwin Miles Stoudenmire on 2020-12-17)
Category:
answer to question

Thank you for the detailed report! The suggestions for how to improve the submission are excellent, and we would be happy to implement them. Here are answers to some of the report's questions that can be covered in a few sentences:

It is indeed correct that $B$ in Eqs (2) and (3) is an arbitrary matrix of its size, and that the projection in Eqs (4) and (5) can be applied more generally than in the context of gradients.

As pointed out, there is a gauge freedom in choosing $W_\perp$, namely unitary rotations on the right index. In practice, what is kept in memory most of the time is $Z = W_\perp B$. When $W_\perp$ and $B$ are needed separately, this can be done by QR decomposing the block matrix $[ W Z ]$ to define $[ W W_\perp ]$. The relevant source code can be found in the TensorKitManifolds.jl package.

Regarding the output logs when running the benchmarks: The Julia scripts simply write to stdout, and the makefile then pipes (or rather tees) the output to the log files. We will improve the README to clarify this, and the other points the report raises.

---

## Round 3 · Referee Report · Anonymous (Referee 2) · 2020-12-31

Strengths

1- Elegant and self-contained summary of the geometry of tensor networks
2- Addresses an important problem in using tensor networks.
3- Links directly to optimization techniques used in broader scientific community and emphasises relationship to existing tensor network optimisation techniques.
4- Clear demonstrations of the (often superior) performance of the new algorithms.

Weaknesses

1- A slightly simpler presentation special cases might be helpful for the non-expert.

Report

This paper harnesses a commanding overview of the geometry of tensor network manifolds to apply state of the art gradient-based optimisation techniques. It brings together these two threads within a self-contained summary of the underlying structures and ideas. The resulting algorithms show state-of-the-art performance. Two example applications are given together with a detailed discussion of how these reduce to previously known algorithms in appropriate limits.

The brevity of the description might make the article a slightly tricky read for those less familiar with the subjects. However, unpacking the ideas further would sacrifice some of the appealing elegance and clarity of the presentation. The balance is about right. I expect it to become a canonical reference on the optimisation of isometric tensor networks.

Requested changes

1- It seems that Eqs. (9) and (10) can also be written using right multiplication of [W,W_\perp] by exp( \alpha [{A,-B^\dagger},{B,0}] ). I find this a little simpler. It is just right multiplication by a unitary. In the Grassmannian case with A=0 the unitary is restricted to preserve the MPS gauge. Is it the case that one can right multiply by an arbitrary unitary on Stiefel manifold?

2- The description of pre-conditioning could be expanded a little for those less familiar with the concept.

3- The authors make a detailed comparison with Evenbly-Vidal polar decomposition-based optimisation methods. It may be useful to spend a few lines making comparison to related schemes such as Rotoflex [Ostaszewski, Grant, and Benedetti, arXiv:1905.09692] that bring quantum circuits to local optima. Such schemes are increasingly being applied in proposals for quantum simulation and the authors will doubtless have useful insights about how these might be improved upon in quantum circuit realisations of MPS.

  • validity: top
  • significance: top
  • originality: high
  • clarity: top
  • formatting: perfect
  • grammar: perfect

Author:  Markus Hauru  on 2021-01-12  [id 1141]

(in reply to Report 2 on 2020-12-31)
Category:
answer to question
pointer to related literature

Thank you for the report! Here are our answers to the questions and proposed changes:

1 - Eqs. (9) and (10) can indeed be written in a different manner as the referee proposes. While this would slightly simplify (9), we find that it would make (10) a bit more complicated and make the similarity between (9) and (10) a little less obvious, and would hence prefer the current formulation. Regarding the question about Stiefel manifolds, yes, for an arbitary unitary $U$, $ \begin{bmatrix}W& W_\perp \end{bmatrix} U \begin{bmatrix} 1 \\ 0 \end{bmatrix}$ is a point on the Stiefel manifold, and any point can be written in this form. Note though that the $\begin{bmatrix} 1 \\ 0 \end{bmatrix}$ part truncates away the last $n - p$ columns ($W$ being $n \times p$), and thus there's a lot of degeneracy in this description, since the last $n - p$ columns of $U$ are irrelevant.

2 - We are happy to accommodate this change.

3 - We thank the referee for pointing out this reference, of which we were unaware. The Rotosolve algorithm seems to be specific for optimising a one-parameter family of unitary gates which are parameterised as the exponential of a single Pauli operator, or a string of Pauli operators, or more generally even, any hermitian matrix (up to an imaginary unit) that is idempotent (i.e. with only eigenvalues are +1 and -1). The cost function is then a periodic function of the coefficient in that exponential, i.e. the angle of rotation or evolution. The algorithm exploits that the full cost function along this line, where all other parameters are fixed, can then be obtained with three measurements, from which the exact minimum can be calculated. The Rotoselect algorithm adds to this the ability to reduce the number of measurements when also wanted to choose from a discrete set of possible generators.

We do not think that these algorithms are directly applicable to the optimisation of MERA, where fully generic 2-qubit (or qu-d-it) gates are required. Such a generic gate could be approximated by a universal quantum circuit made out of gates from the aforementioned type, but this goes beyond our study. The algorithm is reminiscent of a more general construction in the numerical optimisation literature to perform a line search on a manifold of unitary or isometric matrices. When updating a gate along a given direction (determined by the gradient descent, conjugate gradient or L-BFGS algorithm), the problem is also to perform a line search for a one-parameter family of unitaries (given by the chosen retraction). In general though, this will be like the exponential of a hermitian generator (times imaginary unit), but where the generator has several different eigenvalues. In that case, the cost function along the line is not periodic, but possibly quasi-periodic, if the different eigenvalues are close to being harmonic. A line search algorithm based on a Fourier decomposition of the cost function was described in Ref [*] . We have not tested this approach, as this would lead us too far. Furthermore, the line search that we employ (based on an algorithm by Hager and Zhang) is sufficiently efficient. In the context of gradient descent or conjugate gradient, it typically converges (as in satisfying the Wolfe conditions) in two or three evaluations. With L-BFGS, the additional scaling is such that a single evaluation is typically sufficient.

[*] Traian Abrudan, Jan Eriksson, Visa Koivunen, Conjugate gradient algorithm for optimization under unitary matrix constraint, Signal Processing 89 (2009) 1704–1714

---

## Round 4 · Referee Report · Edwin Miles Stoudenmire · 2021-1-23

Strengths

Based on my previous review, and the authors' response and modifications to the paper, I now recommend it for publication. The new material explaining and motivating preconditioning is helpful and welcome, including the pedagogical illustration.

Report

I recommend this paper for publication in SciPost Physics.

---

## Round 4 · Referee Report · Anonymous · 2021-1-29

Strengths

Elegant and authoritative summary of how to best use isometric structure in the optimisation of tensor networks.

Report

The authors have made a number of revisions to the manuscript in response to the first round of review. The additional discussion about the role of preconditioning, in particular, is very helpful.

The authors declined make modifications in response to point 1. I still think that there are advantages to writing the transformation as right multiplication of [W,W_\perp] by the appropriate unitary. The unitary structure combines the MPS left orthogonalization and left tangent gauge choice, and parallel transport expressed as right multiplication by a unitary explicitly preserves these two gauge choices. The parallel transport is also then explicitly independent of the point on the manifold from which it is performed. Both of these points are hidden a little in the present form of (9) and (10). However, this was mainly an aesthetic suggestion and it is acceptable that the authors decide not to follow it.

This is an informative and useful paper and I recommend it for publication.

---

## Editorial Decision

published